

# Salt taste after bariatric surgery and weight loss in obese persons

Cem Ekmekcioglu[1], Julia Maedge[1,2], Linda Lam[1,2], Gerhard Blasche[1], Soheila Shakeri-Leidenmühler[3], Michael Kundi[1], Bernhard Ludvik[4], Felix B. Langer[3], Gerhard Prager[3], Karin Schindler[5] and Klaus Dürrschmid[6]

[1] Institute of Environmental Health, Centre for Public Health, Medical University of Vienna, Vienna, Austria
[2] Department of Nutritional Sciences, University of Vienna, Vienna, Austria
[3] Department of Surgery, Division of General Surgery, Medical University of Vienna, Vienna, Austria
[4] Karl Landsteiner Institute for Obesity and Metabolic Diseases, 1. Medical Department, Rudolfstiftung Hospital Vienna, Vienna, Austria
[5] Department of Internal Medicine III, Division of Endocrinology and Metabolism, Medical University of Vienna, Vienna, Austria
[6] Department of Food Science and Technology, University of Natural Resources and Life Sciences, Vienna, Austria

## ABSTRACT

**Background.** Little is known about the perception of salty taste in obese patients, especially after bariatric surgery. Therefore, the aim of this study was to analyse possible differences in salt detection thresholds and preferences for foods differing in salt content in obese persons before and after bariatric surgery with weight loss compared to non-obese individuals.

**Methods.** Sodium chloride detection thresholds and liking for cream soups with different salt concentrations were studied with established tests. Moreover, a brief salt food questionnaire was assessed to identify the usage and awareness of salt in food.

**Results.** The results showed similar mean sodium chloride detection thresholds between non-obese and obese participants. After bariatric surgery a non-significant increase in the salt detection threshold was observed in the obese patients (mean ± SD: $0.44 \pm 0.24$ g NaCl/L before OP vs. $0.64 \pm 0.47$ g NaCl/L after OP, $p = 0.069$). Cream soup liking between controls and obese patients were not significantly different. However, significant sex specific differences were detected with the tested women not liking the soups ($p < 0.001$). Results from the food questionnaire were similar between the groups.

**Conclusion.** No differences between non-obese persons and obese patients were shown regarding the salt detection threshold. However, due to highly significant differences in soup liking, sex should be taken into consideration when conducting similar sensory studies.

Corresponding author
Cem Ekmekcioglu,
cem.ekmekcioglu@meduniwien.ac.at

## INTRODUCTION

Reaching epidemic dimensions a high body mass index (BMI) belongs to the top risk factors for disability adjusted life years worldwide (*Lim et al., 2012*). Beside environmental, social and genetic factors, the main cause of obesity is a positive energy balance with an

increased consumption of energy dense foods high in sugar and fat, in combination with a lack of physical activity.

In addition to a variety of well known co-morbidities, obese people tend to also have different taste and smell perceptions, which can influence their choice of food (*Drewnowski*, *1997*). Obese children for example identified taste qualities like salty, umami and bitter less precisely compared to children and adolescents of normal weight (*Overberg et al.*, *2012*). Also overweight/obese adults perceived sweet and salty tastes as less intense (−23% and −19%, respectively) than normal-weight controls (*Sartor et al.*, *2011*). On the other hand, *Pasquet et al.* (*2007*) described that massively obese adolescents had higher taste sensitivity than lean controls, especially for sucrose and salt. Furthermore, not only humans but also obese mice show an altered perception of taste (*Maliphol, Garth & Medler*, *2013*).

In previous investigations alterations in taste perception were shown after bariatric surgery and patients found certain foods repulsive and had developed aversions (*Tichansky, Boughter Jr & Madan* , *2006*). Obese women showed decreased cravings for fast food and sweets and decreased preference for high sucrose concentrations after bariatric surgery and weight loss (*Pepino et al.*, *2014*). However, no alterations in taste sensitivity for sweet, salty and savory stimuli were detected.

Former studies especially investigated a change in sweet taste perception of obese people after bariatric surgery, whereas other studies detected an upregulation in bitter and sour taste acuity after surgery (*Bueter et al.*, *2011*; *Scruggs, Buffington & Cowan Jr*, *1994*). Regarding bariatric surgery and salt perception, only few studies are available, with inconsistent results (*Scruggs, Buffington & Cowan Jr*, *1994*; *Pepino et al.*, *2014*).

Due to the sparse and inconsistent data, this study aims to clarify the effect of bariatric surgery on salty taste by analysing the differences in the taste perception of salt of obese persons before and after bariatric surgery in comparison with a non-obese control group. Measurement of salt detection thresholds and analysis of hedonic liking of soups were applied to evaluate physiological and practical aspects.

The relevance of the study is especially twofold. First, a potential variation of salt taste in obesity and after weight loss is of chemosensory and physiological interest. In addition, it is well known that a high salt intake is associated with hypertension (*Ekmekcioglu, Blasche, & Dorner*, *2013*), but probably also obesity (*Ma, He & Macgregor*, *2015*). Therefore a modified salt taste in obese individuals, before or after weight loss, could influence their salt eating behavior and in turn their blood pressure and energy intake.

## MATERIALS AND METHODS

### Participants

A total of 33 obese patients undergoing bariatric surgery (Roux-en-Y- or omega loop gastric bypass) were recruited on a voluntary basis at the Vienna General Hospital (Table 1). Physical examination was performed before and after bariatric surgery and the health status was documented. The test took place one day before patients underwent bariatric surgery and three months after surgery. However, only 19 participants (58%) attended the second test due to personal and organizational issues. Age, sex distribution and BMI of the drop

**Table 1  Characterization of the study groups.**

| Group | Gender | Age (years, mean ± SD) | BMI (kg/m², mean ± SD) | BMI, 3 months post OP (mean ± SD, kg/m²) | Major comorbidities |
|---|---|---|---|---|---|
| Controls ($n = 29$) | Female: 14 | 41.0 ± 12.8 (range: 22–62) | 23.6 ± 3.0 (range: 16–28) | n.a. | No indicated |
| Patients ($n = 33$; post OP: $n = 19$) | Female: 21 | 46.3 ± 10.0 (range: 23–65) | 43.2 ± 5.7 (range: 30–51) | 33.8 ± 5.0 (range: 22–42) | Hypertension: 17 Diabetes: 10 |

**Notes.**
n.a., not applicable.

outs were not significantly different from the group who participated at the second test. The control group consisted of healthy persons with a BMI between 16 and 28 (Table 1). The sex distribution between the controls and the patients was not significantly different as analysed by a $\chi^2$-test.

In relation to previous similar studies (*Pepino et al.*, *2014*; *Bueter et al.*, *2011*; *Scruggs, Buffington & Cowan Jr*, *1994*; *Burge et al.*, *1995*) the sample size of the present study can be regarded as sufficient, having enough power to detect significant effects.

Informed written consent was obtained by all participants and the project was approved by the ethics committee of the Medical University of Vienna (EK Nr.: 1193/2013).

## Sensory evaluation

Salty taste detection thresholds were assessed with the "Standard Practice for Determination of Odor and Taste Thresholds By a Forced-Choice Ascending Concentration Series Method of Limits", which is a standard practice developed by ASTM International (*American Society for Testing and Materials*, *2011*). The three alternative forced choices (3 AFC) test was used to identify the threshold of the participants (*Peng, Jaeger & Hautus*, *2012*). In eight blocks increasing sodium chloride solutions (∼0.003 to ∼0.034 mol/L or ∼0.16 g/l to ∼2 g/L; ISO 3972 Standard) were tested. Each block consisted of two stimuli of distilled water (20 ml) and one sodium chloride stimulus (20 ml) in increasing concentrations. The test samples were labelled with random three-digit numbers, so that the participants were not able to identify the plastic vessels with the salt. After tasting each block the study persons had to indicate the sample that tasted different. They rinsed their mouth with tap water before tasting each block.

Based on the correct/incorrect responses of each participants an individual best-estimated threshold was calculated. An example for two participants is shown in Table 2.

The best-estimate threshold (BET) for test person number 1 is: $\sqrt{0.48 x 0.69} = 0.58$ g NaCl/L. The BET for test person number 2 is: $\sqrt{0.69 x 0.98} = 0.82$ g NaCl/L.

## Assessment of the preference for soups differing in salt concentrations

Cream soups with increasing salt concentrations were prepared with tap water, sour cream, and flour. The six NaCl concentrations of soups varied from 0.051 mol/L (∼3 g/L) to 0.154 (∼9 g/L) in 0.026 mol/L steps. For the production of one liter of soup, 35 g of wheat

**Table 2  Examples of the salt threshold measurement.**

| Participants | Judgments | | | | | | | | Best-estimate threshold (BET) (g NaCl/L) |
|---|---|---|---|---|---|---|---|---|---|
| | NaCl concentrations increase → (g NaCl/L) | | | | | | | | |
| | 0.16 | 0.24 | 0.34 | 0.48 | 0.69 | 0.98 | 1.4 | 2 | |
| 1 | 0 | 0 | 0 | 0 | 1 | 1 | 1 | 1 | 0.58 |
| 2 | 1 | 0 | 1 | 0 | 0 | 1 | 1 | 1 | 0.82 |

**Table 3  Liking scores for cream soups.**

| 9-point hedonic scale | | |
|---|---|---|
| 9 | Like extremely | Liking area |
| 8 | Like very much | |
| 7 | Like moderately | |
| 6 | Like slightly | |
| 5 | Neither like nor dislike | Neutral |
| 4 | Dislike slightly | Dislike area |
| 3 | Dislike moderately | |
| 2 | Dislike very much | |
| 1 | Dislike extremely | |

flour (brand "Clever") and one cup of sour cream (250 g, brand "milfina," 15% fat) were blended and 250 ml of tap water was added to the flour-sour cream mixture and stirred together. By using a funnel the mixture was filled in a 1 L volumetric flask. Residues in the bottle were dissolved with water and transferred into the volumetric flask until all visible residues were removed. The bottle was then filled up with water to 1 L and was shaken intensively. Afterwards the content was divided into five 200 ml glass bottles for heating in a stirring block thermostat. The bottles were closed throughout the entire cooking process to prevent the evaporation of the soup. To avoid agglutination or burning of the ingredients at the bottom of the bottles a magnetic stirrer was used during cooking. The different concentrations of salt (brand "Bad Ischler Tafelsalz") were added after the soup reached the temperature of 60 °C and the content of the bottles were stirred again. After cooking, the soups were poured into 20 ml plastic cups, closed with numbered plastic lids and deep frozen at −20 °C. For the sensory tests portion of soup were reheated in a microwave to 50 °C ± 1 just before testing.

A 9-point hedonic scale, with intervals from 1 (dislike extremely) to 9 (like extremely), was used to assess the preference of the participants' liking for the five different soups (Table 3).

## Food questionnaire

A food questionnaire which included eight questions about the preference for salty foods, use of salt in the preparation of meals, and awareness for salt in foods was completed by the study participants.

The questions were:

- I like to eat.... (unsalted to very salty).. meals
- I..... (never to always)... add salt to my meals
- How much salt do you add into water, when cooking two portions of spaghetti (approx. 300 g)?
- How frequently do you eat ham, sausages or bacon?
- How frequently do you eat convenience products (for ex. pizza, packet soups, deep frozen dishes)?
- How often do you eat salty nuts, chips, popcorn or similar salty snacks?
- Which of the listed foods below (sausages, raw meat, uncooked spaghettis, bread, milk, cheese, fresh vegetables, nuts, convenience foods, and chocolate) are high or low in salt?
- Do you think that dietary salt has negative/positive/no effects on our health?

The questions were taken from previous studies addressing the eating or salt behavior of adults and partly modified according to Austrian eating habits and conditions (*Bundesministerium für Verbraucherschutz, Ernährung und Landwirtschaft, 2008*; *Kim et al., 2007*; *Parmenter & Wardle, 1999*; *Webster et al., 2010*).

## Statistics

Differences in the mean values of estimated thresholds between patients and controls were analysed with a Mann–Whitney-$U$-test, since the data were not normally distributed. Comparison between estimated thresholds before and after surgery in the patient group was done by a Wilcoxon-test.

Soup liking between patients vs. controls was analysed with an univariate analysis of variance with the weighted mean of the soup liking (soup 1 was weighted with 1 and soup 5 with 5) as the dependent variable, group and sex as factors and age and BMI before surgery as covariates.

Differences in soup liking before and after surgery in the patients were calculated by an analysis of repeated measures with age, sex and the difference in BMI (before and after surgery) as covariates. Tests for inner subject contrasts were used to analyse differences between the soups.

Data from the food questionnaire were compared with an unpaired $t$-test between the two groups.

The statistical analysis was performed on IBM SPSS 20 Statistics with $p$ values of less than 0.05 being regarded as significant.

## RESULTS

### Estimated salt detection thresholds

No statistical difference was observed between the salt detection thresholds of the controls (mean $\pm$ SD: 0.49 $\pm$ 0.45 g NaCl/l) compared to the obese group (mean $\pm$ SD: 0.51 $\pm$ 0.34 g NaCl/l) (Fig. 1A). Furthermore, a non-significant increase in the salt detection threshold of the patients was observed after bariatric surgery (mean $\pm$ SD: 0.44 $\pm$ 0.24 g NaCl/L before OP; 0.64 $\pm$ 0.47 g NaCl/L after OP, $p = 0.069$) (Fig. 1B). No sex specific differences were

A

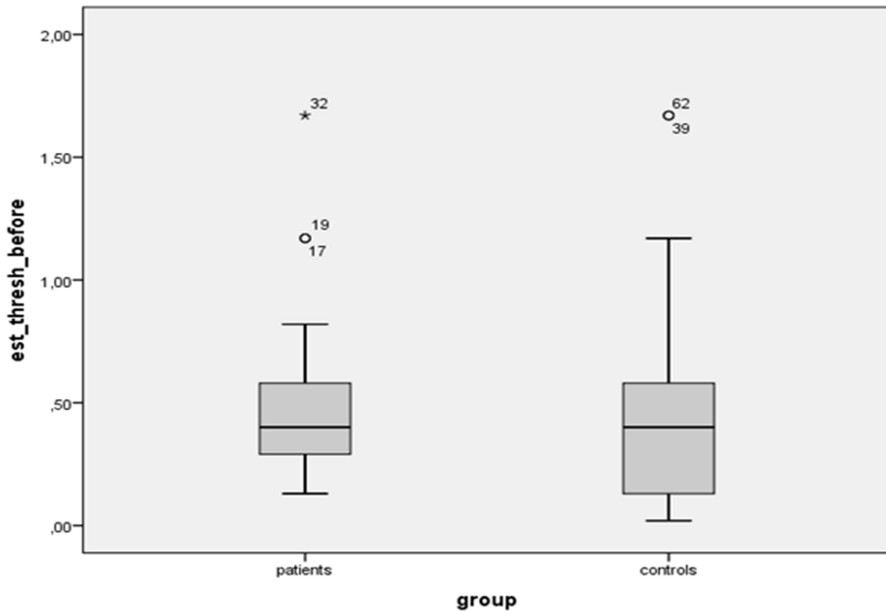

B

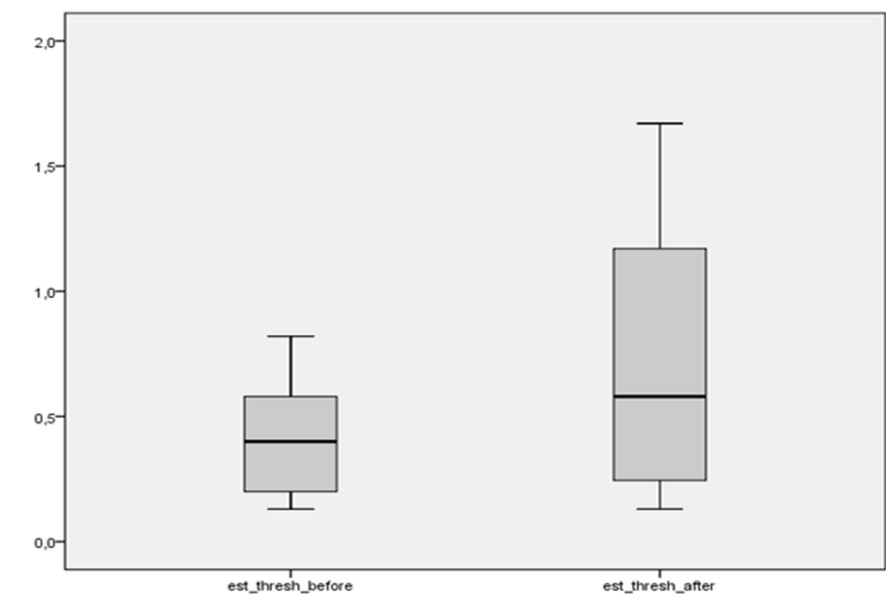

**Figure 1 Estimated salt detection threshold between patients before and after bariatric surgery and controls.** (A) Estimated salt detection threshold (in g NaCl/L) between patients before bariatric surgery ($n = 33$) and controls ($n = 29$). Mean values were not significantly different. (B) Estimated salt detection threshold (in g NaCl/L) between patients before vs. after bariatric surgery (both $n = 19$). Mean values were not significantly different.

**Table 4** Liking scores of soups 1–5 in controls and patients before and after OP.

| Group | Soup 1 | Soup 2 | Soup 3 | Soup 4 | Soup 5 |
|---|---|---|---|---|---|
| Controls ($n = 29$) | $3.59 \pm 2.31$ | $4.24 \pm 2.33$ | $4.14 \pm 2.55$ | $3.45 \pm 2.41$ | $2.93 \pm 2.20$ |
| Patients before surgery ($n = 33$) | $2.79 \pm 2.04$ | $2.79 \pm 2.38$ | $2.88 \pm 2.36$ | $3.42 \pm 2.66$ | $2.88 \pm 2.66$ |
| Patients after surgery ($n = 19$) | $2.89 \pm 1.82$ | $2.58 \pm 1.90$ | $2.05 \pm 1.68$ | $2.21 \pm 1.96$ | $1.74 \pm 1.41$ |

**Notes.**
Values are shown as mean $\pm$ SD.

detected. Furthermore there were no significant differences in the salt detection thresholds between patients with or without diabetes or hypertension.

## Soup liking

Cream soups with five different salt concentrations in increasing order (soup 1 lowest, soup 5 highest salt concentration) were used to test the soup preference on a 9 point hedonic scale. The results showed that the control group liked the soup 2 best ($4.24 \pm 2.33$), whereas patients before surgery preferred soup 4 most (Table 4).

However statistical analysis showed no significant differences between controls and patients before surgery when using the weighted mean of the soup liking as the dependent variable (Fig. 2A).

Interestingly, we found highly significant sex specific differences in the soup liking ($p < 0.001$; Fig. 2B), suggesting that the tested women did not like the soups.

In addition, the soup liking before and after surgery were not significantly different in the patient group. Also, contrast analyses showed no significant differences between the soups.

## Food questionnaire

The food questionnaire was completed by 15 controls and 26 obese patients after testing the salt detection thresholds and soup liking at the first visit. Except for the question asking for the salt content of chocolate no significant differences were found between the two groups. Major results from the questionnaire showed that the participants in both groups ate moderately salted foods, rarely added salt to cooked dishes, ate ham, sausages or bacon 1–2 times per week, and ate convenience products or salty chips and similar snacks 2–3 times per month (for data, please refer to Supplemental Information 1).

## DISCUSSION

The present study showed that the salt detection thresholds and soup liking of obese patients were not significantly different from healthy, non-obese controls. In addition, a non-significant increase in the salt detection threshold was observed in patients after bariatric surgery. An unexpected result was that the tested women did not like the cream soups.

In several studies, the association between taste perception and obesity was evaluated, with a special focus on sweet taste (reviewed in *Donaldson et al.*, *2009*). However, the results

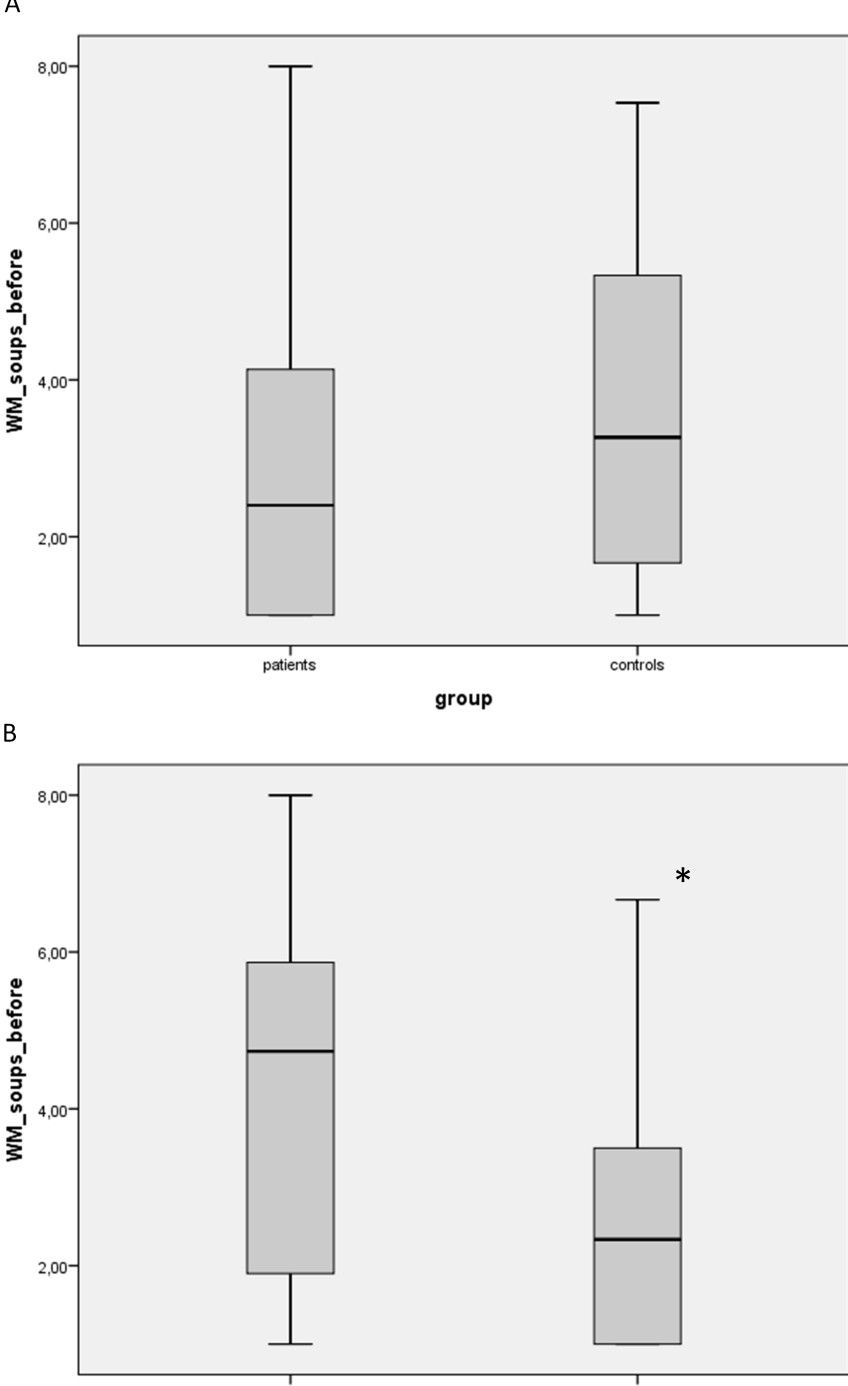

**Figure 2** **Soup liking between patients (as the weighted mean, WM) vs. controls (A) and also sex specific effects (B) were analysed with a univariate analysis of variance with the weighted mean of the soup liking (soup 1 was weighted with 1 and soup 5 with 5) as the dependent variable, group and sex as factors and age and BMI before surgery as covariates.** (A) no statistical difference between patients vs. controls. (B) *significantly lower than males ($p < 0.001$).

in these studies were partly contradictory, possibly because different techniques were used (*Donaldson et al.*, *2009*).

Regarding obesity surgery and taste, one of the first studies in the mid 1970s showed that one year after intestinal bypass surgery and significant weight loss obese patients rated the pleasantness of a 40% sucrose solution lower than before surgery (*Bray et al.*, *1976*). In another study, the recognition threshold for sucrose fell from 0.047 mol/L to 0.019 mol/L 12 weeks after Roux-en-Y gastric bypass (*Burge et al.*, *1995*).

Also Bueter et al. investigated changes of sweet taste perception in nine obese patients 2 months after gastric bypass surgery. They found that even though patients were able to detected lower concentrations of sucrose after the operation as compared to before, no differences in hedonic ratings of sucrose solutions were found (*Bueter et al.*, *2011*). Therefore, although measurements of taste detection thresholds are established physiological methods to study the function of sensory receptors and gustatory circuits they do not necessarily correlate with suprathreshold intensity or hedonic values (*Bueter et al.*, *2011*; *Spector*, *2000*; *Bartoshuk*, *1978*; *Webb et al.*, *2015*). For example, in a recent comparison of five methods to assess taste function it was shown that detection thresholds were not correlated with suprathreshold intensities of any taste quality, including sodium chloride (*Webb et al.*, *2015*).

Regarding salty taste and obesity, only a few studies are available. In a study from *Pasquet et al.* (*2007*) for example, lower recognition thresholds for sucrose and sodium chloride were found in obese adolescents compared to the lean control group. In another study it was shown that overweight women liked the taste of salty foods more compared to the normal weight group; however converse results were seen in the male control group (*Donaldson et al.*, *2009*). Furthermore, Japanese women declaring to like salty foods had a higher BMI compared to those who disliked them (*Hashimoto et al.*, *2008*). Also, results from the French web-based observational cohort of the Nutrinet-Santé study showed that overall liking scores for salt and fat were linearly positively related to BMI in 46,909 adults and were higher in obese than in normal-weight persons (*Deglaire et al.*, *2015*). However, no effect of BMI on salty taste perceptions was shown in other studies (*Malcolm et al.*, *1980*; *Simchen et al.*, *2006*).

Salt detection threshold after bariatric surgery was only addressed by few studies. In rats, no effect of gastric bypass on salt preference scores was found (*Bueter et al.*, *2011*). Similar to our study, *Scruggs, Buffington & Cowan Jr*, (*1994*) also found no significant differences in the salt detection threshold between morbid obese patients compared to lean controls. However, in some contrast to our study the salt detection threshold 30, 60 or 90 days after bariatric surgery declined non-significantly, indicating a trend for an increased sensitivity. One major reason for this discrepancy may be the very low sample size of 6 individuals in the study of *Scruggs, Buffington & Cowan Jr* (*1994*). Furthermore, Scruggs et al. used higher salt concentrations and a different method for the tests, all which could have explained the diverging results. In another recent study, *Pepino et al.* (*2014*) showed no effect of bariatric surgery and weight loss on sodium chloride detection threshold in 27 obese women.

We also studied the liking of cream soups differing in their salt concentrations, since taste thresholds may be less representative for daily life. We found no significant differences

between the patients and the controls and no effect of bariatric surgery and weight loss on hedonic responses. However, interestingly, considerable sex dependent differences were detected, as the women in our study did not like the cream soups. We also analysed soup liking between controls and patients and before and after surgery in women and men separately, and, similar to the outcome in the whole group, in both cases the results were not significant.

The reasons for the unexpected lower soup liking scores in women compared to men are unclear. Possibly lower intakes of milk and milk products in women may be an explanation. This was for example shown in a large sample by *Klesges et al.* (*1999*) with non-Hispanic white women experiencing more gastric stress after milk consumption than men. In addition, gender specific differences in liking of sweet, fatty or salty foods (*Deglaire et al.*, *2015*) or taste acuity (*Mojet, Heidema, & Christ-Hazelhof*, *2003*) may also be relevant. However, to the best of our knowledge we found no studies looking at sex differences in the liking of dairy products. Furthermore, it has been suggested that the taste of women may be affected by the various phases of the menstrual cycle (*Verma et al.*, *2005*). A further reason for the sex specific differences may be a more frequent dieting behavior in women which may have lead to avoidance or disliking of cream soups by (dieting) women. In conclusion, further studies are needed to confirm these results.

Previous studies suggested an association between BMI and liking for salt (*Deglaire et al.*, *2015*; *Donaldson et al.*, *2009*) and it has been shown that higher urinary sodium excretions, as an indicator of salt intake, is associated with increased body weight (*Huh et al.*, *2015*; *Libuda, Kersting & Alexy*, *2012*). Therefore, we also evaluated the salt eating behavior of the participants with a short questionnaire and found no differences between obese and lean persons in the preference for use and awareness of salt in food.

One limitation of the study was that the salt intake of the participants was not evaluated. In this regard a recent study found an association between salt intake and salty taste sensibility among hypertensive and normotensive individuals (*Piovesana Pde, Sampaio Kde & Gallani*, *2013*). However, since there were no differences between the usage of salt between patients and controls, we believe that there might be no relevant differences in the salt intake between the study groups. Another limitation is the relatively high dropout rate of the patients after bariatric surgery resulting in a lower statistical power in the analysis of the effects of weight loss on the salt detection threshold. Finally, although we additionally studied soup liking, we did not analysed systematically suprathreshold intensity ratings, as an additional way to evaluate hedonic components of higher levels of salt stimuli. This is also a drawback of this study. In general, in a previous study it was suggested that taste function is very complex and difficult to characterize (*Webb et al.*, *2015*).

In conclusion, in our study obesity was not associated with an altered salty taste or salting of a taste soup. Bariatric surgery with weight loss may tend to worsen the salt detection threshold of obese patients, although studies with higher sample sizes are necessary. Furthermore, sex should be considered when conducting sensory studies.

## ACKNOWLEDGEMENTS

The authors like to thank Prof. Regina Sommer and her team from the Water Hygiene Unit, Institute for Hygiene and Applied Immunology, Medical University of Vienna, for technical support.

### Funding
The authors received no funding for this work.

### Competing Interests
The authors declare there are competing interests.

### Author Contributions
- Cem Ekmekcioglu conceived and designed the experiments, analyzed the data, wrote the paper, prepared figures and/or tables, reviewed drafts of the paper.
- Julia Maedge and Linda Lam conceived and designed the experiments, performed the experiments, prepared figures and/or tables, reviewed drafts of the paper.
- Gerhard Blasche analyzed the data, reviewed drafts of the paper.
- Soheila Shakeri-Leidenmühler, Bernhard Ludvik, Felix B. Langer and Gerhard Prager contributed reagents/materials/analysis tools.
- Michael Kundi analyzed the data.
- Karin Schindler conceived and designed the experiments, reviewed drafts of the paper.
- Klaus Dürrschmid conceived and designed the experiments, contributed reagents/materials/analysis tools, reviewed drafts of the paper.

### Human Ethics
The following information was supplied relating to ethical approvals (i.e., approving body and any reference numbers):

The project was approved by the ethics committee of the Medical University of Vienna (EK Nr.: 1193/2013).

### Data Availability
The raw data has been supplied as Supplemental Information 1.

### Supplemental Information
Supplemental information for this article can be found online at http://dx.doi.org/10.7717/peerj.2086#supplemental-information.

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
