# Peer review of "Salt taste after bariatric surgery and weight loss in obese persons"

_PeerJ, doi:10.7717/peerj.2086_

## Round 0.1 · original submission · Major Revisions

Please carefully consider the comments raised by the reviewers, specially those regarding methodological aspects of the study.

Please also note the annotated manuscript provided by reviewer 1.

Reviewer 1 ·

Basic reporting

I find no areas where the article fails to meet PeerJ standards.
Some language editing is required, and decimal points replace commas throughout the MS. Other comments in the attached PDF.

Experimental design

The experimental design is appropriate for measuring potential threshold changes and differences.
However, the variability in the soup-liking estimations pretty much makes this measure useless for interpretation of results. Authors might have considered a wider range of sodium concentrations.

Validity of the findings

The threshold findings are of interest.
The soup findings are problematical, cf above.

Additional comments

Authors should address the difference between threshold and suprathreshold measures of taste. There are findings relating threshold and preference and intake, but I believe the preponderance of the evidence dissociates the two.1) they should distinguish in their introduction, discussion, and referencing between the two - they are not the same! 2) They should consider this difference in reference to their chosen methodology. I understand that they may have done so (without adequate rationalisation or explanation) in including the soup test, but its weakness undermines the interest in the report. This issue is central to the import of the paper, namely, whether the surgery alters the preference, and potentially intake, of food in relation to its sodium content.This might be an important issue for the health and dietary guidance of these patients. This paper is. unfortunately, limited in contributing insight to the issue.
Nevertheless, I would encourage the authors to address the issues appropriately so that the threshold data will be available to guide further research.

Annotated reviews are not available for download in order to protect the identity of reviewers who chose to remain anonymous.

·

Basic reporting

Introduction lacks information on the importance of this study. Why is altered salt taste is important in patients after RYGB? How is this going to aid weight loss or weight loss maintenance?

Experimental design

Line 103/104 were the tastants prepared using tab water or distilled water? i.e. was the mineral contents of tab water considered in the preparation of the tastants?

Validity of the findings

No comments

Additional comments

Line 118 Please consider changing ‘table water’ to ‘tab water’

Line 176 increased grams of salt means decrease detection sensitivity hence higher detection thresholds, please consider changing it to ‘Furthermore, a non-significant increase in the salt detection threshold of the patients was observed after bariatric surgery’. This was explained correctly in line 206.

Table 3 Please give example of salt level in a typical type of soup or cream soup

---

## Round 0.2 · accepted · Accept

I think the manuscript has improved with the modifications included by the authors. Now, the diferent aspects raisen by the reviewers are clearer. Therefore, the manuscript is now acceptable for publication.